# Bone Marrow Endothelial Cells Increase Prostate Cancer Cell Apoptosis in 3D Triculture Model of Reactive Stroma

**DOI:** 10.3390/biology11091271

**Published:** 2022-08-26

**Authors:** Lindsey K. Sablatura, Tristen V. Tellman, Aemin Kim, Mary C. Farach-Carson

**Affiliations:** 1Department of BioSciences, Rice University, 6100 Main St., Houston, TX 77005, USA; 2Department of Diagnostic & Biomedical Sciences, University of Texas Health Science Center at Houston School of Dentistry, 7500 Cambridge St. Room 4401, Houston, TX 77054, USA

**Keywords:** bone marrow tumor microenvironment, triculture system, apoptosis, endothelial cells, prostate cancer

## Abstract

**Simple Summary:**

Prostate cancer (PCa) metastasizes preferentially to the bone marrow where it becomes difficult to treat. PCa cells in the bone marrow may survive, dormant and undetected for many years before patients eventually relapse with metastatic disease. Bone marrow is a complex tissue that initially is hostile to the PCa cells, Understanding how cancer cells survive in the bone marrow and what changes to the bone microenvironment permit them to switch to an actively growing state could offer new therapeutic strategies to combat metastatic PCa. In this study, we describe a method to culture PCa cells with two other cell types from the bone marrow, stromal cells and endothelial cells, as a way to study the interactions among these cell types. We found that factors produced by bone marrow endothelial cells, but not endothelial cells from other tissues, trigger PCa cells to either die or enter a dormant state, similar to what has been observed in patients when PCa cells initially colonize the bone marrow. Further analysis of the cell interactions within the culture model described in this study will offer increased understanding of PCa interaction with the bone marrow environment.

**Abstract:**

The bone marrow tumor microenvironment (BMTE) is a complex network of cells, extracellular matrix, and sequestered signaling factors that initially act as a hostile environment for disseminating tumor cells (DTCs) from the cancerous prostate. Three-dimensional (3D) culture systems offer an opportunity to better model these complex interactions in reactive stroma, providing contextual behaviors for cancer cells, stromal cells, and endothelial cells. Using a new system designed for the triculture of osteoblastic prostate cancer (PCa) cells, stromal cells, and microvascular endothelial cells, we uncovered a context-specific pro-apoptotic effect of endothelial cells of the bone marrow different from those derived from the lung or dermis. The paracrine nature of this effect was demonstrated by observations that conditioned medium from bone marrow endothelial cells, but not from dermal or lung endothelial cells, led to PCa cell death in microtumors grown in 3D BMTE-simulating hydrogels. Analysis of the phosphoproteome by reverse phase protein analysis (RPPA) of PCa cells treated with conditioned media from different endothelial cells identified the differential regulation of pathways involved in proliferation, cell cycle regulation, and apoptosis. The findings from the RPPA were validated by western blotting for representative signaling factors identified, including forkhead box M1 (FOXM1; proliferation factor), pRb (cell cycle regulator), and Smac/DIABLO (pro-apoptosis) among treatment conditions. The 3D model presented here thus presents an accurate model to study the influence of the reactive BMTE, including stromal and endothelial cells, on the adaptive behaviors of cancer cells modeling DTCs at sites of bone metastasis. These findings in 3D culture systems can lead to a better understanding of the real-time interactions among cells present in reactive stroma than is possible using animal models.

## 1. Introduction

Once metastatic, disseminated tumor cells (DTCs) preferentially colonize specific organs, such as bone marrow in the case of prostate cancer (PCa) metastasis, because those organs harbor a fertile environment for metastatic growth. The bone marrow tumor microenvironment (BMTE) consists of a dense network of collagen-rich extracellular matrix and cells, including immune, stromal, and endothelial cells, which provide paracrine and juxtacrine signaling factors [1]. While seemingly a rich growth environment, it can take years or decades after surgical resection or radiation of the primary tumor for metastases to become clinically overt [2], suggesting that bone marrow is not initially a permissive microenvironment for epithelial prostatic tumors. Bone marrow reactive stromal cells produce factors including interleukin 6 (IL-6), transforming growth factor beta (TGFβ), and tumor necrosis factor alpha (TNF-α) that induce apoptosis and neuroendocrine differentiation in PCa cells [3,4] and activate the transcription of matrix proteins such as perlecan [5] and tenascin-C [6] in the reactive stroma. Current thinking acknowledges the adaptive changes that occur in the bone metastatic niche that allow DTCs to survive, transdifferentiate, and eventually thrive in the bone marrow environment, eventually leading to widespread bone metastases and, too often, lethal disease [3,7].

In a study seeking to understand site-specific influences on PCa growth in xenograft mouse models of metastasis, it was observed that the majority of PCa cells injected directly into the bone marrow became dormant or died, while those injected subcutaneously proliferated to form large tumor masses [8]. This study reinforced the idea of an initially hostile bone marrow environment to DTCs but did not identify the cell source of the factors controlling cancer cell growth in bone marrow. Elucidating the cell source(s) and paracrine-activated pathways that control DTC apoptosis, dormancy, and reactivation is critical for the identification of new therapeutic targets to eliminate DTCs before they gain a foothold in the bone niche or to prevent the reactivation of dormant cells to stop the formation of overt bone metastases and prevent disease progression.

Murine models of cell death and dormancy, while high in biological relevance, are time-consuming, expensive, and often difficult to experimentally control [9]. Additionally, real-time interactions between cells in the reactive microenvironment of the tumor cannot be readily observed in living animals. Complex 3D culture models allow a reductionist reconstitution of the BMTE that enables facile manipulation of cell populations, paracrine factors, and extracellular matrix properties in a platform that allows for high-resolution imaging. This study describes the construction of a 3D hyaluronic acid (HA)-based in vitro triculture model and its application to examine the influence of bone marrow stromal and endothelial cells in a model of prostate DTCs in the BMTE. In this model, we utilized various cell lines that recapitulate the metastatic BMTE, including PCa cells representing different subtypes (C4-2B, PC-3, and 22Rv1), non-hematopoietic bone marrow stromal cells (HS27a), and microvascular endothelial cells from different tissues (BMEC-1, HMEC-1, and HULEC-5a). C4-2B, PC-3, and 22Rv1 cells represent three different subtypes of PCa [10] that all are derived from bone metastatic tumors. The HS27a cell line is derived from bone marrow stroma [11] and is a rich source of TGFβ in the 3D triculture model [12]. The BMEC-1, HMEC-1, and HULEC-5a cell lines in this model represent the microvascular endothelial cell population. Each cell line provides the context-specific influence of the endothelium in various tissues, where BMEC-1 comes from the bone marrow, HMEC is derived from the dermal microvasculature, and HULEC-5a cells come from the lung endothelium. The construction of 3D systems containing various combinations of these cells and/or conditioned media from them in co-culture allows us to examine the real-time interactions among these cell types in reactive stroma that might account for the differences in tumor growth observed when prostate DTCs metastasize to various organ sites.

## 2. Materials and Methods

### 2.1. Cell Culture

Human bone marrow stromal cells (HS27a), human dermal microvascular endothelial cells (HMEC-1), human lung microvascular endothelial cells (HULEC-5a), and PC3 PCa cells were purchased from the American Type Culture Collection (Manassas, VA, USA). Bone marrow microvascular endothelial cells (BMEC-1) [13] were a gift from Dr. Carlton Cooper (University of Delaware), 22Rv1 prostate cancer cells [14] were a gift from Dr. Neil Bhowmick (Cedars Sinai, Los Angeles), and C4-2B PCa cells [15] were a gift from Dr. Leland Chung (at that time at the University of Texas MD Anderson Cancer Center). HS27a, 22Rv1, PC3, and C4-2B were cultured in RPMI 1640 (Gibco, Invitrogen, Carlsbad, CA, USA)) supplemented with 10% (*v*/*v*) heat-inactivated fetal bovine serum (Atlanta Biologicals, Inc., Flowery Branch, GA, USA) and 1X penicillin–streptomycin (Gibco). HMEC-1 and HULEC-5a were cultured in MCDB131 (Gibco) supplemented with 10% (*v*/*v*) heat-inactivated fetal bovine serum, 10 mM L-glutamine (Gibco), 10 ng/mL Epidermal Growth Factor (Gibco), and 1 µg/mL hydrocortisone (Sigma-Aldrich, St. Louis, MO, USA). BMEC-1 cells were cultured in Medium 199 (HEPES, no L-glutamine from Sigma) supplemented with 15% (*v*/*v*) heat-inactivated fetal bovine serum, 10 mM L-glutamine, 1X penicillin–streptomycin, 40 ug/mL endothelial cell growth supplement (ECGS, Sigma), and 16 U/mL heparin (Sigma).

### 2.2. HA Hydrogel Components

Thiolated hyaluronan (Glycosil^®^) and polyethylene glycol diacrylate (PEGDA) MW3400 (Extralink^®^) were purchased from BioTime Inc., now Advanced BioMatrix, Carlsbad, CA, USA. Protease-cleavable peptide, KGGGPQGIWGQGK (PQ peptide), with N-terminal acetylation (GenScript, Piscataway, NJ, USA) was reacted with acrylate-PEG-SVA, MW3400 (Laysan Bio Inc., Arab, AL, USA) at a molar ratio of 1:2.5 in HEPBS buffer (20 mM HEPBS, 100 mM NaCl, and 2 mM MgCl_2_, pH 8.0). The reaction was allowed to proceed overnight at 4 °C with shaking and was protected from light. The reaction solution was then dialyzed in a 3500 Da MWCO dialysis membrane (Spectrum LifeSciences, LLC, Rancho Dominguez, CA, USA) against ultrapure water for 2 days, filter-sterilized, frozen at −80 °C, and lyophilized for 48 h. Integrin-binding peptide GRGDS with C-terminal amidation (GenScript) and laminin peptide YIGSR (GenScript) were reacted with acrylate-PEG-SVA at a molar ratio of 1.2:1 using the same protocol. PEGylation was verified by MALDI-TOF (Bruker AutoFlex II). The lyophilized powder was stored at −20 °C and allowed to come to room temperature before use.

### 2.3. Mono-, Co-, and Triculture Encapsulation

Molds for hydrogel gelation were custom-made as described previously [16]. Briefly, 1.5 mm thick slabs of PDMS were laser cut into 24 × 60 mm rectangles containing multiple 6 mm diameter circular holes. These molds were cleaned, autoclave-sterilized, and press-sealed onto sterile glass slides to form well cavities with approximately 50 µL volume.

Thiolated hyaluronan (HA-SH, Glycosil^®^) was reconstituted according to the manufacturer’s instructions. HA-SH solution pH was adjusted to 8.0 with 1N NaOH immediately before use. Cells to be encapsulated were harvested with trypsin EDTA (0.125% for BMEC-1 and 0.25% otherwise, from Gibco) and counted.

Cells were resuspended in HA-SH solution, and then acrylate-PEG-GRGDS (73.7 mg/mL in PBS), acrylate-PEG-YIGSR (73.7 mg/mL), and acrylate-PEG-PQ-PEG-acrylate (37.0 mg/mL in PBS) were added at a volume ratio of 4:0.5:0.5:0.2:0.8 (HA-SH:acrylate-PEG-GRGDS:acrylate-PEG-YIGSR:acrylate-PEG-PQ-PEG-acrylate:PBS) for all experiments. The solution was mixed well, and then 42 µL of gel solution was dispensed into each well cavity of the PDMS mold, as described previously [17]. Filled molds were placed into the cell culture incubator at 37 °C for 45 min to allow gelation to occur; then, one drop of cell culture media was added to the top of each gel to prevent dehydration, and the molds were placed back into the incubator for 1–2 h. This allowed low-crosslinking-density hydrogels to crosslink more completely for ease of handling before the gels were scored around their circumference with the tip of a sterile needle. Next, the PDMS mold was removed, and the hydrogels were transferred with a sterile spatula to a 48-well culture plate with 400 µL of media per well. Co-culture media consisted of endothelial cell basal media (Medium 199 or MCDB131) in a 1:1 (*v*/*v*) ratio with DMEM (high glucose, Gibco) supplemented with 2% (*v*/*v*) heat-inactivated fetal bovine serum, 10 mM L-glutamine, 1X penicillin–streptomycin, 40 µg/mL endothelial cell growth supplement, and 16 U/mL heparin. 

For triculture and conditioned media experiments, PCa cells were pre-clustered using microwell plates (Sphericalplate 5D, Kugelmeiers, Erlenbach, Switzerland) by dispensing 90,000 cells per well of a 24-well plate, as described previously [18]. Cells were allowed to cluster for 24 h before collecting and encapsulating them.

### 2.4. Conditioned Media Experiments

Populations of endothelial and cancer cells were maintained in 2D culture at 40–90% confluency in the relevant co-culture media. Conditioned media were removed from 2D cultures every 3 days, centrifuged to remove cells and debris, and mixed 1:1 with fresh co-culture media to feed PCa hydrogel cultures. Because of the potential effect of nutrient depletion in medium conditioned by cells, we chose to use a homologous (i.e., cancer cell-conditioned) medium rather than fresh medium as the control for the endothelial cell conditioned media. 

### 2.5. Live/Dead Assay

To assess cell viability, the media were removed, and hydrogels were rinsed with HBSS. HBSS containing 1 µM calcein AM (Biotium, Freemont, CA, USA), 1 µM Hoechst 33342 (Enzo Life Sciences, Inc., Farmingdale, NY, USA), and 3 µM ethidium homodimer-1 (Biotium) was added to the hydrogels at a volume of 250 µL per well. After 1 h of incubation, Z-series images were obtained using a Nikon A1-R confocal microscope with 20× objective, resonance scanning, 2× averaging to reduce noise, and a Z-step size of less than 1 µm. 

### 2.6. Immunostaining

Hydrogels were fixed for 10 min with paraformaldehyde (4%, *w*/*v*) in phosphate buffer with calcium and magnesium and then rinsed with HBSS. Cells were permeabilized for 10 min with Triton X-100 (0.2%, *v*/*v*) in PBS at room temperature, then blocked with goat serum (3%, *v*/*v* in 0.2%, *v*/*v*, Triton X-100 in PBS) for 1 h, and incubated with primary antibody in blocking solution overnight at 4 °C. The cultures were then rinsed 3 × 10 min with PBS, incubated for 2 h with secondary antibody in blocking solution, rinsed 3 × 10 min with PBS, and counterstained with DAPI (3 µg/mL). Stained cultures were imaged as described in Section 2.5 but with 8× averaging to reduce noise for the comparatively dimmer immunofluorescence signal.

### 2.7. TUNEL Assay

C4-2B cells were seeded at a density of 100,000 cells/well in microwell plates. Cells were cultured in the microwell plates to retain spherical cultures, and conditioned medium mixed at a ratio of 1:2 (conditioned media:fresh C4-2B complete media) was exchanged every 2 days. Cell clusters were fixed on days 3, 7, and 14 with paraformaldehyde (4%, *w*/*v*). Clusters were then stained using the Promega DeadEnd™ Colorimetric TUNEL system using the standard protocol with appropriate modifications. Between steps, clusters were allowed to settle to the bottom of the 1.5 mL tube via gravity, and the supernatant was removed. Following the manufacturer’s protocol, clusters were allowed to develop until light brown and then washed with de-ionized water. Clusters were stored in PBS at 4 °C and imaged using the BZ-X810 Keyence with the 10× objective.

### 2.8. Cell Labeling with Fluorescent Proteins

BMEC and HS27a cells were stably labeled with GFP and RFP, respectively, using pre-made lentiviral particles from Amsbio, Cambridge, MA, USA (LVP426 and LVP429 with FP expression under an EF1α promoter and puromycin selection).

### 2.9. Image Analysis

Confocal Z-stack live/dead/nuclei images were analyzed with Imaris (Oxford Instruments, Abingdon, UK) using the Cells and Spots algorithms. The Spots algorithm identifies objects as a point or “spot” based on a predicted diameter. The Cells algorithm creates a volumetric surface object with one channel (for example, green calcein AM signal) and then identifies spots inside of it using a different channel (for example, blue Hoechst signal). Viability (%) is the number of living cells (identified using the Cells algorithm on the calcein and Hoechst channels) divided by the sum of the number of living cells plus the number of dead cells (identified using the Spots algorithm on the EthD channel) multiplied by 100. Cluster diameter was approximated for irregularly shaped, non-spherical objects by averaging the ellipsoid axis lengths (A, B, and C, representing radii) of each calcein object and multiplying by 2. The distance between green endothelial structures and red stromal structures was determined using the Distance Transform function to identify the distance of the closest point of each red (stromal) object to any green (endothelial) structure. Direct contact is defined by this distance equaling zero.

TUNEL images were analyzed using ilastik version 1.4.0b27 with the pixel classification + object classification feature [19]. Individual JPEGs were loaded into the software. Feature selection values were color/intensity σ_0_ = 0.30, edge σ_1_ = 0.70, and texture σ_1_ = 0.70. Cluster area was trained using label 1 for the entire cluster length and label for the background values. For stain area, label 1 designated the stain signal, and label 2 designated background cluster values. In object feature selection, only size in pixels under standard object features was used as a measure. Desired objects were selected in object classification, and size in pixels for each object was exported into Excel. 

### 2.10. Reverse Phase Protein Array Preparation

Samples for the reverse phase protein array (RPPA) were prepared as per the instructions of the University of Texas M.D. Anderson Cancer Center RPPA Proteomics Facility. Pre-clustered C4-2B cells were cultured as they were in the conditioned media experiments, with 50,000 cells per gel. On day 14, 8–10 gels were pooled into a single sample and centrifuged to pellet the cells from the now-very-soft gels. The supernatant was removed, and the cell pellet was resuspended in 150 µL of cold lysis buffer (1% Triton X-100, 50 mM HEPES, pH 7.4, 150 mM NaCl, 1.5 mM MgCl_2_, 1 mM EGTA, 100 mM NaF, 10 mM Na pyrophosphate, 1 mM Na_3_VO_4_, and 10% glycerol, containing freshly added protease and phosphatase inhibitors). Samples were incubated for 20 min on ice with brief vortexing every 5 min and then centrifuged at 14,000 rpm for 10 min at 4 °C. The supernatant was collected, protein concentration was assayed, and samples were diluted to 1.5 µg/µL in lysis buffer. Samples were mixed 3 parts lysate to 1 part sample buffer (40% Glycerol, 8% SDS, and 0.25 M Tris-HCL, pH 6.8, with 10% (*v*/*v*) freshly added β-mercaptoethanol), boiled for 5 min, and then stored at −80 °C until submission to the core. All samples were submitted in duplicate.

The RPPA methodology can be found at the University of Texas M.D. Anderson Cancer Center RPPA proteomics core website (https://www.mdanderson.org/research/research-resources/core-facilities/functional-proteomics-rppa-core.html, accessed on 5 June 2022) and has been described in detail previously [20,21]. Samples were probed with 472 unique antibodies (see antibody list at (https://www.mdanderson.org/research/research-resources/core-facilities/functional-proteomics-rppa-core/antibody-information-and-protocols.html, accessed on 5 June 2022, Set 174)).

### 2.11. Western Blotting

Conditioned media-treated cells were lysed with 500 µL of RIPA buffer (Thermo Scientific 89900) with 1X HALT Protease Inhibitor (Thermo Scientific 78430, Waltham, MA, USA). Cell lysates were transferred to 1.5 mL centrifuge tubes and further broken down by shearing with an insulin syringe (Exel International 26027, Redondo Beach, CA, USA). Homogenized lysates were then assayed for total protein content via BCA protein assay (Thermo Scientific 23227). A total of 20 µg of each sample was transferred to a clean centrifuge tube and mixed with 5X SDS-containing loading buffer with β-mercaptoethanol. Samples were heated at 100 °C for 5 min and then centrifuged briefly in a table-top centrifuge. Denatured samples were loaded on a 10% Bis-Tris protein gel (Invitrogen NP0301BOX, Waltham, MA, USA) in a MOPS SDS running buffer-filled electrophoresis system (Invitrogen NP0001) and run at 150 V for 1 h. Gels were transferred to nitrocellulose membranes at 50 V for 4 h in tris-glycine transfer buffer containing 20% (*v*/*v*) methanol. Membranes were blocked in 3% (*w*/*v*) bovine serum albumin resuspended in TBSt on an orbital shaker for 1 h at room temperature. Blots were incubated with primary antibodies (FOXM1: Santa Cruz sc-271746; pRb: Cell Signaling Technology 9308; and Smac/DIABLO: Cell Signaling Technology 2954) at a 1:3000 dilution in 3% (*w*/*v*) bovine serum albumin in TBSt overnight at 4 °C on an orbital shaker. Blots were then washed 3× for 5 min in TBSt and then incubated at room temperature for 1 h with HRP-conjugated secondary antibody (Jackson Immuno Research, West Grove, PA, USA) at a 1:50,000 dilution. Blots were washed 3 × 10 min, then incubated for 1 min with substrate (Pierce 32106), and developed on film. Densitometry of western blot bands was performed using ImageJ software. To determine relative protein levels, values for FOXM1, pRb, or Smac were first normalized to the GAPDH loading control. Normalized values then were ratiometrically compared to the C4-2B CM 24hr treatment (control) group, which was set to 1.0.

## 3. Results

### 3.1. Endothelial–Stromal Co-Cultures

An endothelial-to-stromal cell ratio of 4:1 was originally chosen based on previous work performed by the West lab [22]. Varying this ratio within the physiological endothelial-cell-to-pericyte ratio range between 1:1 and 10:1 [23] (8:1, 4:1, or 2:1 ratios with 100,000 cells total) showed that higher stromal cell content (2:1) prevented the formation of interconnected networks of EC tubules (Figure 1). Imaris analysis using the Filament Tracer algorithm indicated that the total length of tubule structures was similar for 8:1 and 4:1 BMEC to HS27a ratios on day 14 (Figure 1d). The 8:1 cultures formed more intricately connected, spidery networks, as indicated by the number of branch points (Figure 1c), while the networks formed in 4:1 cultures were more robust. HS27a closely associated with BMECs, with 50–60% of HS27a cells in direct, pericyte-like contact with endothelial structures in all three BMEC:HS27a ratio cultures by day 14 (Figure 1b). Higher crosslinking density—as determined by the ratio of HA-SH to acrylate-PEG-PQ-PEG-acrylate—was found to slow endothelial network formation (Appendix A). The 15:1 thiol-to-acrylate ratio used for the studies described formed gels that were sturdy enough to be handled easily but also porous enough to allow tubule formation within 2 weeks.

### 3.2. Endothelial–Stromal–Cancer Triculture

Mixed tricultures were established by encapsulating 40,000 BMEC-GFP, 10,000 HS27a-RFP, and 25,000 pre-clustered C4-2B. C4-2B cells were pre-clustered into ~100 µm tumoroids using a microwell plate with the rationale that clustered PCa cells would proliferate in triculture when single cells in previous triculture experiments did not. Cultures were fixed and C4-2B cells were labeled by immunofluorescence for EpCAM. BMEC + HS27a (BH) co-cultures behaved as expected, with the majority of the organization occurring in the second week of culture (Figure 2). Cancer cells in C4-2B + HS27a (CH) cultures proliferated over time. HS27a cells in CH cultures very quickly migrated to and infiltrated the cancer clusters. Surprisingly, in BMEC + HS27a + C4-2B tricultures (BHC), many cancer cells died off by day 7, leaving only small clusters of healthy cells by day 14. This death/dormancy response was accompanied by a much larger increase in red signal (HS27a) within cancer clusters compared to CH cultures, suggesting increased HS27a migration toward C4-2B in triculture.

### 3.3. Effect of Conditioned Media from Bone Marrow, Lung, and Dermal Endothelial Cells

Apoptosis-inducing factors can be secreted via paracrine or juxtacrine signals. To examine the role of paracrine signaling in the decline of C4-2B PCa cells in triculture, 25,000 pre-clustered C4-2B cells were encapsulated per hydrogel. PCa hydrogels were cultured in 1:1 (*v*/*v*) fresh media + conditioned media from BMEC-1 cells (B-CM) or from C4-2B cells (C-CM) as a control for the exhaustion of nutrients. Another group of hydrogels was also treated with conditioned media from BMEC-1 cells that had been previously treated with conditioned media from C4-2B cells (CtoB-CM) to account for the possibility of BMEC-1 requiring pre-stimulation from PCa cells to produce dormancy/death-inducing factors. C4-2B clusters in all groups proliferated normally until day 7 (Figure 3a). Clusters cultured in either media condition containing B-CM began to decline in numbers between days 7 and 14 to the point at which the B-CM and CtoB-CM conditions contained about half the cell population of the C-CM cultures on day 14 (Figure 3b). This decrease in cell number was accompanied by a decrease in percent viability for those same conditions (Figure 3c).

To investigate the tissue specificity of the PCa cell death response, C4-2B cells were cultured in conditioned media from the microvascular endothelium of one of two other microenvironmental contexts: lung, to which PCa also commonly metastasizes, and skin, to which it does not. C4-2B cells cultured in HMEC-1 dermal microvascular endothelial cell-conditioned medium (D-CM) and C4-2B cultured in C-CM grew normally until day 7 and then maintained the same total number of cells and nearly 100% viability until day 14 (Figure 3d and Appendix A). This contrasts with the result of B-CM treatment, in which both the total cell number and percent viability declined between day 7 and day 14. HULEC-5a lung microvascular endothelial cell-conditioned media (L-CM) also had no effect on C4-2B growth or viability (Figure 3e and Appendix A).

To determine whether the response observed in C4-2B cells treated with B-CM is specific to this cancer cell line, pre-clustered 22Rv1 or PC3 PCa cells were treated with B-CM. 22Rv1 cells cultured in B-CM- or 22Rv1-conditioned media (R-CM) showed the same trend in growth and viability regardless of the conditioned media treatment (Figure 3f and Appendix A). B-CM also showed no effect on the growth of PC3 cells compared to cells grown in PC3-conditioned media (P-CM) (Figure 3g and Appendix A).

B-CM- and C-CM-treated cultures were fixed and immunostained with antibodies for Ki67 and cleaved Casp3 as proliferative and apoptotic markers, respectively. B-CM hydrogels on day 14 showed a clear increase in the percentage of cells with the cleaved Casp3 signal, calculated by comparing the number of red objects to the number of nuclei (Figure 4a,b). The Ki67+ population increased in both conditions over time, with a slightly, but not significantly, larger increase in C-CM (Figure 4c). The Ki67/Casp3 ratio of day 14 B-CM cultures approached one, indicating a steady-state population, with proliferation balancing apoptosis (Figure 4d). The Ki67/Casp3 ratio of C-CM cultures on day 14 was much higher than 1, as these cells were still in an actively growing state.

A TUNEL assay of C4-2B cells treated for 3, 7, and 14 days showed the effects of conditioned medium from C4-2B, HS27A, and BMEC cells across time (Figure 5a,b). On day 3, there was little difference in size or levels of apoptosis between different treatment groups. By day 7, however, C4-2B clusters differed greatly in size and levels of apoptosis, with the groups containing BMEC-conditioned media showing the smallest clusters overall and the clusters with the highest level of apoptosis. By day 14, cluster size between the C4-2B- and HS27A-conditioned media groups showed little difference, but the groups with BMEC-conditioned media showed a high level of apoptosis and resulting size and shape differences (Figure 5a,b), indicating the specific role of BMEC-conditioned media in the induction of an apoptotic phenotype in cell clusters.

### 3.4. RPPA Assay and Target Validation

The next series of experiments sought to identify the downstream signaling pathways leading to the increase in apoptosis and reduction in viability in PCa cells cultured with endothelial cells. Treatment of C4-2B PCa cells with B-CM or H-CM and subsequent analysis via RPPA to examine the phosphoproteome in treated PCa cells showed a distinct signature specific to B-CM factors not seen with D-CM- or C-CM-treated cells (for full results of the RPPA; see Appendix A). Among the pathways examined, several, including those involved in cell cycle regulation, apoptosis, autophagy, and proliferation, were strikingly differently affected by B-CM than by treatment with D-CM or C-CM (Figure 6a). Representative molecular targets from these pathways were chosen for validation by western blot. For these studies, the C4-2B parent cell line, C4-2, was used to represent changes from the pre-bone-adapted PCa cell, which is the best cell model in this series for the DTC arriving in the BMTE. Pro-apoptotic Smac/DIABLO levels were elevated in B-CM-treated C4-2 cells as compared to D-CM or C-CM controls by 24 h (Figure 6b and Appendix A). The cell cycle regulator pRb (Ser807/811) was decreased in cells treated with B-CM when compared to controls at all time points (Figure 6c and Appendix A). Forkhead box M1 (FOXM1), associated with cell proliferation (Figure 6d and Appendix A), showed decreased levels at 24, 48, and 72 h specific to B-CM-treated groups, also consistent with RPPA data. Additionally, markers of cellular senescence CDKN2/p16^ink4a^ and CDKN1A/p21 did not change between treatment groups in the RPPA analysis, while AXL, a known dormancy initiator in the context of PCa bone metastasis, increased in cancer cells treated with B-CM in the RPPA analysis. Our data suggest that PCa clusters treated with B-CM contain a mixture of apoptotic and dormant cells and may reflect a similar situation for DTCs in bone.

## 4. Discussion and Conclusions

The BMTE to which PCa DTCs home is a complex system of multiple interacting cell types, extracellular matrix proteins, and regulatory factors and is very different from the primary tumor microenvironment in the prostate. These differences pose a challenge during the early phases of DTC colonization, requiring the adaptation of surviving cancer cells to a new environment that they initially find to be hostile to growth. Interestingly, PCa DTCs have been reported to co-opt the mechanisms of hematopoietic stem cell homing to and maintenance in bone, where they preferentially adhere to bone marrow endothelium as compared to the endothelium of other organs [24,25]. These findings indicate that a close behavioral relationship exists between PCa DTCs and bone marrow endothelium that may not exist at other metastatic sites.

To study this relationship of reactive stroma in real time, something not possible in animal xenograft models, we constructed a customized 3D hydrogel model supporting the triculture of PCa cells growing as tumoroids with bone marrow endothelial and bone marrow stromal cells. Our observations that the extent of vascular network formation in these hydrogels was highly dependent on cell ratios of endothelial and stromal cells are consistent with earlier studies [22] and demonstrate that cellular cross-talk was occurring even in the absence of PCa cells. Our finding that stromal HS27a cells were closely associated with BMECs in successful networks, with 50–60% of HS27a cells in direct, pericyte-like contact with endothelial structures, provides evidence that bone marrow stromal cells can stabilize microvascular-like endothelial networks in 3D systems. This is consistent with earlier studies [26] demonstrating a key role for pericytes in the tumor microenvironment in vascular survival.

When we moved to the triculture system introducing PCa tumoroids, it was initially surprising that we observed a loss of cellularity in tumoroids along with a death/dormancy phenotype. The induction of PCa cell line, C4-2B, apoptosis by B-CM suggests paracrine signaling through specific BMEC-produced soluble factors not produced by comparable microvascular cells of the lung (HULEC-5a) or dermal (HMEC-1) endothelium. However, it should be noted that apoptosis in the BMTE likely results from the interplay of multiple juxtacrine and paracrine factors, so this observation does not discount the role of cell–cell interactions, such as those demonstrated by the attraction of HS27a stromal cells to C4-2B tumoroids and their subsequent invasion in co-culture. A contextual representation of these cellular interactions can be found in Figure 7. C4-2B cells in triculture also likely attract stromal cells, but the presence of the RFP signal originally from HS27a within EpCAM-positive cancer cells suggests that the fluorescent protein or even the stromal cells themselves could be internalized by the cancer cells in response to pro-apoptotic signals present in cultures containing BMECs [27].

In the reactive BMTE, cancer cells, endothelial cells, and stromal cells co-evolve, a phenomenon that has been called the “bystander effect” [28,29]. An “angiocrine switch” during tumor progression occurs as a consequence of changes in gene expression in the proximal tumor endothelium [30]. Numerous endothelium-derived angiocrine factors are suspected to regulate tumor growth and metastasis, including insulin-like growth factor 1 (IGF-1) [31], IL-6 [32], C-C motif chemokine ligand 5 (CCL5) [33], and others [34]. Bone marrow endothelial cell angiocrine signaling to PCa cells has also been described [35], though it is not well understood which factors are involved. Likewise, the precise mechanisms that determine whether the perivascular niche is quiescence-promoting or pro-proliferative remain elusive. The 3D triculture system created in this work will provide a useful platform for further study of the cellular interactions and the molecular identities of paracrine/juxtacrine factors that regulate PCa cell behavior in the BMTE. In particular, it will be useful to replace established cancer cell lines with patient-derived cells, a goal we have for the future. In previous work, we showed that immune cells can also be grown with PCa tumoroids and bone marrow stromal cells [36], opening the door to ever more complex hydrogel systems to mimic the BMTE for cancer biology studies.

PCa can be categorized into prostate cancer subtypes (PCS), as defined in the Prostate Cancer Transcriptome Atlas (http://www.thepcta.org/ accessed on 5 June 2022). These three major PCS categories are defined by their gene expression patterns, where PCS1 and PCS2 reflect tumors of luminal subtypes and PCS3 represents a more basal subtype of PCa [10]. The C4-2 and C4-2B PCa cells used in this study are representative of the PCS2 subtype, while 22Rv1 PCa cells are of the more aggressive PCS1 subtype, though both are androgen-responsive and primarily osteoblastic in the BMTE. PC3 PCa cells are of the PCS3 subtype and represent a more neuroendocrine phenotype, a characteristic generally seen as a survival mechanism in response to factors produced in the BMTE or to hormonal or chemotherapy [37,38]. PC3 cells are more basal in origin and represent androgen receptor-negative disease. Of note, inherent phenotypic differences among these subtypes may provide some insight into why the PCa cell lines used in this study responded differently to B-CM treatment.

Activation of PCa growth can occur downstream of various cell-surface signaling events through receptors such as plexin-semaphorins, TGFβRIII, and c-Met [8,9,18]. FOXM1, a transcription factor, is known as a master regulator in growing cancers, where it plays an essential role in sustaining proliferation [39]. Rb can play a similar role in cell cycle progression, regulated by its phosphorylation at specific residues. Phosphorylation of Rb at serine 807/811 blocks its ability to bind to Abl, modulating downstream apoptosis signaling [40,41]. Second mitochondria-derived activator of caspase/direct inhibitor of apoptosis-binding protein with low pI (Smac/DIABLO) plays a role in the mediation of apoptotic signaling, where it promotes apoptosis and cell cycle arrest [42]. We observed the differential effects of various conditioned media treatments, first identified by RPPA in C4-2B cells, an osteoblastic bone marrow-adapted cell line, and then validated by western blotting in C4-2, another LNCaP-derived, DTC-like cell line [15].

Regulation of Rb through phosphorylation is complex and not well understood. Phosphorylation of Rb at Ser807/811 is involved in the induction of both cellular senescence [43] and the mitochondrial apoptosis pathway [44]. We did not observe changes in downstream protein activation, such as CDKN2/p16ink4a or CDKN1A/p21 [45], that would be expected during the induction of senescence by Rb (Ser807/811), implying that Rb (Ser807/811) may instead be inducing apoptosis or playing some other role in the context of our endothelial cell-PCa cell interactions. AXL is a receptor tyrosine kinase implicated in promoting cancer cell survival, drug resistance, acquisition of a stem-like phenotype, and tumor progression [46]. The AXL receptor and its ligand growth arrest–specific 6 (GAS6) act to initiate dormancy in the context of bone metastatic PCa [47]. Inhibition of this pathway using AXL-targeted shRNA or a small molecule inhibitor prevented the formation of bone metastases in murine breast and PCa models [48]. GAS6 is expressed by endothelial cells [49], and the increase in AXL expression in cancer cells exposed to BMEC suggests that the GAS6/AXL signaling axis may contribute to the endothelial cell-induced death/dormancy phenotype observed in this study. It is possible that the modulation of these pathways, involved in apoptosis, cell cycle regulation, and proliferation, by endothelial cells in the BMTE is a vital part of the adaptive process that occurs in reactive stroma. In this scenario, the initial death of a majority of DTCs is followed by the quiescence/dormancy of survivors for a variable period, followed by reactivation and eventual return to proliferation and the formation of clinically overt metastases. This idea is consistent with our observation that PCa clusters contain mixtures of apoptotic and dormant cells removed from the cell cycle. Thus, bone marrow endothelial cells appear to play a key role in initially limiting the growth of metastatic PCa cells at bone sites of metastasis, but for patients who develop bone metastases, this inhibition is process-limited in time, and the cancer cells eventually overcome the stromal resistance to form tumors.

Further phenotypic analysis of the residual cancer cells in this model will reveal whether these cells possess characteristics of dormant bone marrow resident DTCs, including quiescence and resistance to therapy-induced apoptosis [50]. Refinement of the model system to enable an extended culture period could also facilitate studies to identify specific factors governing cancer recurrence, which has been reported to occur even in low-risk patients after surgical resection of the primary tumor [51]. Such a model could provide an avenue for developing DTC-targeted therapies [52] that do not rely on costly, time-consuming in vivo models.

This study described a paracrine death or dormancy/quiescence-inducing signaling axis that is specifically mediated by microvascular endothelial cells from bone marrow, not those from other sites, regardless of their status as common sites of PCa metastasis. This interaction is also specific to the more indolent, luminal, PCS2 subtype of PCa, a cell phenotype likely to represent the early DTC. Additionally, our 3D triculture system offers a unique tool to study the impact of the reactive stroma in metastatic PCa in real time, something impossible in animal studies. While further study is needed to elucidate additional players in these pathways, the results shown here could have a profound impact on our understanding of PCa interaction with the BMTE, paving the way for novel, precision medicine approaches to prevent bone metastasis from becoming lethal.

## Figures and Tables

**Figure 1 biology-11-01271-f001:**
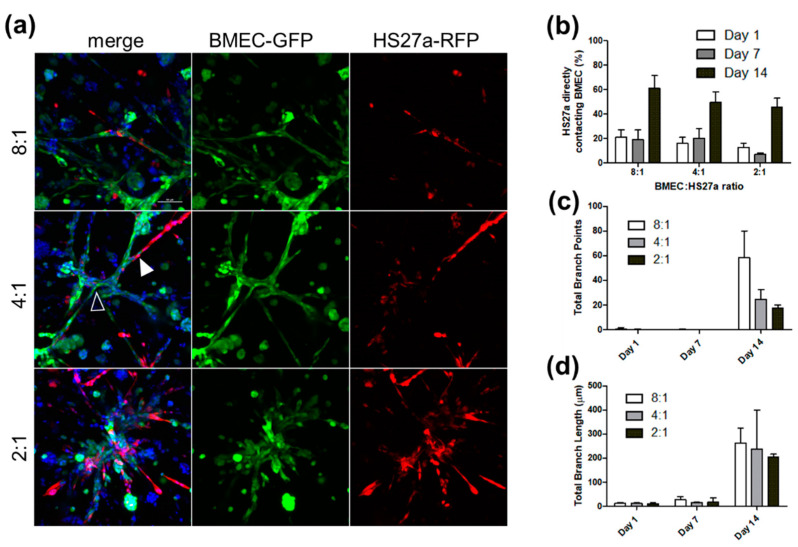
**BMEC-to-HS27a ratio impacts endothelial network architecture.** BMEC-GFP and HS27a-RFP were encapsulated in 8:1, 4:1, and 2:1 ratios, cultured for 14 days, and then labeled with Hoechst and imaged. The results show that 8:1 cultures form more intricately branched networks than 4:1 cultures, while 2:1 cultures show much less interconnectivity. HS27a-RFP is associated with BMEC-GFP structures in all conditions (**a**). The white arrow indicates the pericyte-like association of HS27a with BMEC. The white outlined arrow indicates a branch point. Scale = 100 µm. The number of HS27a directly in contact with BMEC structures increases over time in all BMEC:HS27a ratios (**b**). The total branch length is similar between BMEC:HS27a ratios (**d**), while 8:1 cultures show the largest number of branch points (**c**), indicating a more intricately branched network. Each bar represents nine images across three separate hydrogels. Error bars represent standard deviation.

**Figure 2 biology-11-01271-f002:**
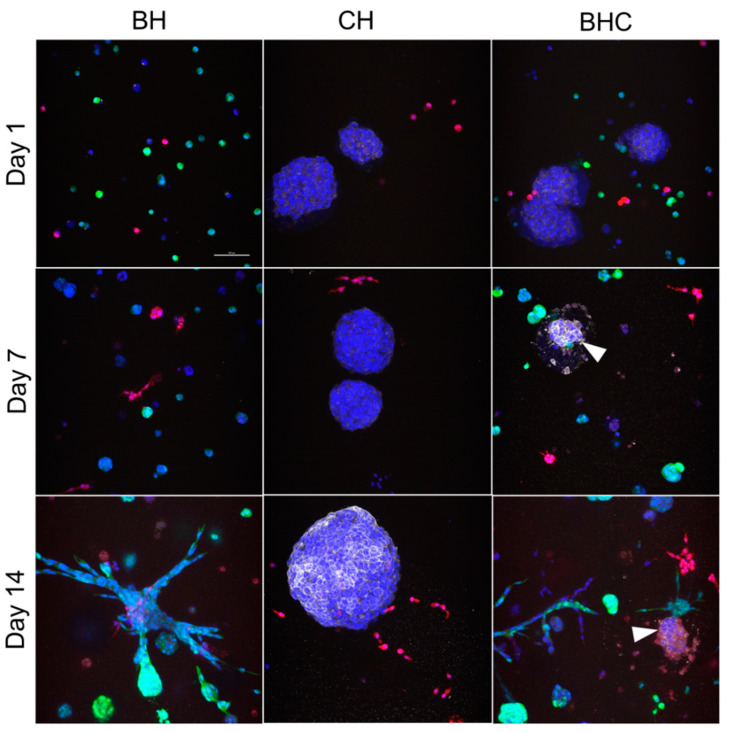
**Triculture of BMEC, HS27a, and pre-clustered C4-2B induces a cell death/dormancy phenotype in C4-2B.** BMEC-GFP (green), HS27a-RFP (red), and C4-2B pre-clustered into ~100 µm tumoroids were encapsulated in co- and tricultures. BMEC + HS27a (BH), C4-2B + HS27a (CH), and BMEC + HS27a + C4-2B (BHC) gels were fixed with 4% paraformaldehyde on days 1, 7, and 14 and stained with EpCAM (white) to label epithelium-derived PCa cells and DAPI (blue) to label nuclei. Scale bar = 100 µm for all figures.

**Figure 3 biology-11-01271-f003:**
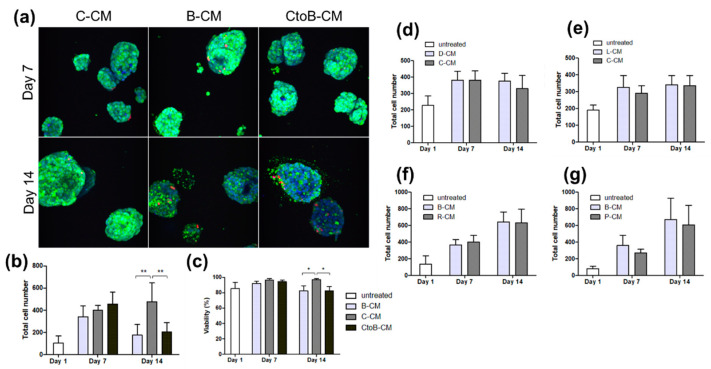
**BMEC-conditioned media have a PCa-cell-type-specific effect on cell viability, not present in ECs of other tissues.** Pre-clustered C4-2B cells were treated with BMEC-conditioned media (B-CM), C4-2B-conditioned media (C-CM), or conditioned media from BMEC cells that were treated with C4-2B-conditioned media (CtoB-CM). Cells were stained with calcein AM, ethidium homodimer, and Hoechst and imaged by confocal microscopy (**a**). IMARIS analysis of confocal z-stacks indicates a decrease in total cell number (**b**) and percent viability (**c**) on day 14 in conditions containing BMEC-conditioned media. Graphs represent means + SD. Statistical significance using two-way ANOVA with Bonferroni posttest, * *p* < 0.05, ** *p* < 0.01. Pre-clustered C4-2B cells were treated with HMEC-1-conditioned media (D-CM) or C4-2B-conditioned media (C-CM) (**d**) and HULEC-5a-conditioned media (L-CM) or C4-2B-conditioned media (C-CM) (**e**). Pre-clustered 22Rv1 (**f**) or PC3 (**g**) cells were treated with BMEC-conditioned media (B-CM) or the conditioned media from 22Rv1 (R-CM) or PC3 (P-CM), respectively.

**Figure 4 biology-11-01271-f004:**
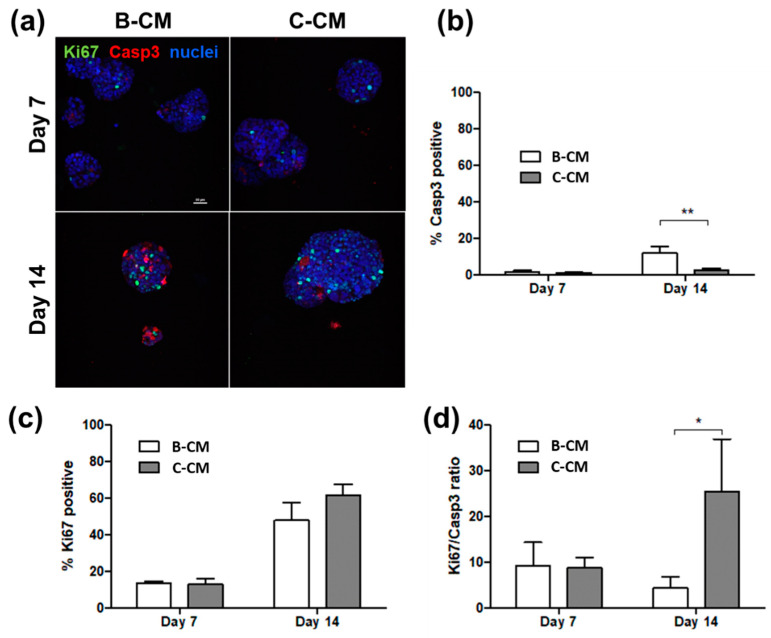
**BMEC-conditioned media increases the number of Casp3+ cells.** B-CM- and C-CM-treated cultures were fixed and stained for Ki67 and cleaved Casp3 (**a**). IMARIS analysis confirms an increase in Casp3+ cells on day 14 in B-CM (**b**), with a slight but not significant decrease in Ki67 (**c**). Accordingly, the Ki67/Casp3 ratio is different between the two conditions, with B-CM Ki67/Casp3 being near 1 (**d**). Scale bar = 50µm. Graphs represent means + SD. Statistical significance using two-tailed Student’s *t*-test, * *p* < 0.05, ** *p* < 0.01.

**Figure 5 biology-11-01271-f005:**
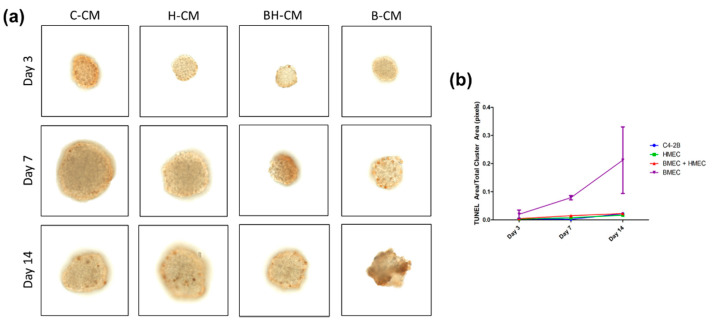
**TUNEL assay of conditioned medium (CM)-treated C4-2B clusters.** Colorimetric TUNEL assay of C4-2B clusters cultured in microwells and treated with cancer cell CM (C-CM), HS27A-only CM (H-CM), BMEC and HS27A CM (BH-CM), and BMEC only CM (B-CM) on days 3, 7, and 14. Dark brown spots indicate cells that are undergoing apoptosis (**a**). Size in the pixel area of stain (dark brown) over total cluster area for each treatment group (**b**).

**Figure 6 biology-11-01271-f006:**
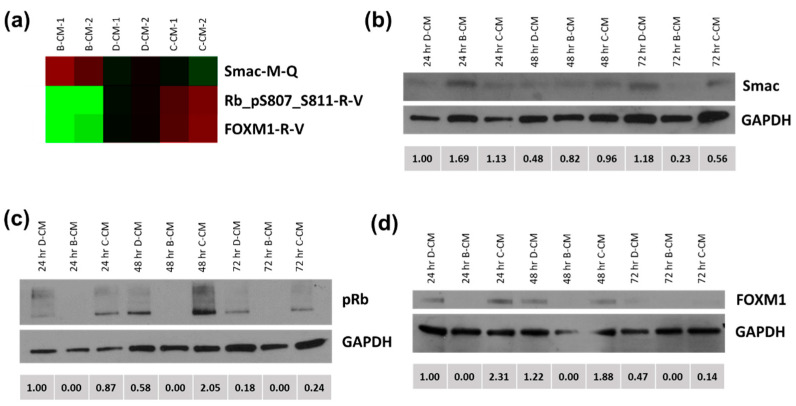
**BMEC-conditioned media selectively suppresses phosphorylation of Rb and expression of Smac and FOXM1 in C4-2 cells.** RPPA analysis of BMEC (B-CM)-, HMEC (D-CM)-, and C4-2 (C-CM)-conditioned medium treatment of C4-2B cells showed differential expression of signaling molecules pRb, Smac, and FOXM1 (downregulated proteins = green; upregulated proteins = red) (**a**). B-CM treatment showed differential expression of Smac (**b**), pRb (**c**), and FOXM1 (**d**) across time and between control treatment groups (D-CM and C-CM) quantified via densitometry below each corresponding blot.

**Figure 7 biology-11-01271-f007:**
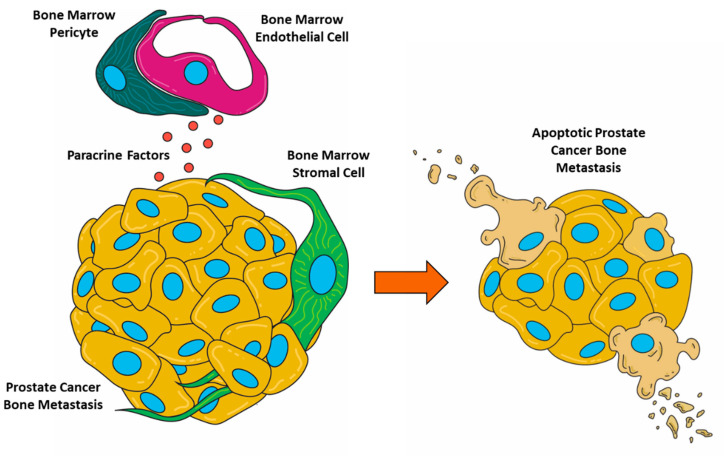
Graphical representation of cell interactions in the BMTE.

## Data Availability

Not applicable.

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
