# Peer review of "Bone Marrow Endothelial Cells Increase Prostate Cancer Cell Apoptosis in 3D Triculture Model of Reactive Stroma"

_biology, 2022, doi:10.3390/biology11091271_

Round 1
Reviewer 1 Report (New Reviewer)
This manuscript describes the establishment of a complex 3D/triculture system. They show that BMEC suppress PrCa cells that are reminiscent of DTC (cell lines PC3, 22Rv1 and C24B). (lung- and dermal EC do not have this effect on PrCa cells.
The model is elegant, and the quantification is well structured.
The authors conclude that there is a suppressing/quiescence inducing effect of the BMEC, which could explain the well described long dormancy of PrCa cells in the bone marrow. The conclusion would benefit if the authors would speculate on how the model could help in understand the transition from dormancy to progression of the DTCs.
Minimal; abstract l 220 should be endothelial cells (in stead of epithelial cells)
Author Response
We thank the reviewer for the enthusiasm expressed for our manuscript.
We changed epithelial to endothelial in the abstract (thanks for the good catch!).
We added a short paragraph to the Discussion in which we speculate on the ways that this co-culture model might help in the development of therapies targeting DTCs.
Reviewer 2 Report (New Reviewer)
Sablatura et al. report a 3D model to investigate the effect of reactive BMTE, including stromal and endothelial cells, on the adaptive behavior of cancer cells simulating DTC at the site of bone metastasis. The topic of this article is of interest. Overall, the interaction between different cell types was effectively simulated by modeling different subtypes of prostate cancer cell lines. However, organoid cell clusters are more easily formed due to cell lines in vitro, the study is relatively preliminary. Human tumor samples or organoids were used, which would add rigor to the study. In addition, previous studies have shown that DTC is dormant in bone marrow. What is the state of residual tumor cells after co-culture in this 3D model? Is it in a dormant state? How resistant is it to apoptosis? And how is this resistance achieved? Authors should verify and discuss. Residual DTC is the most important component that should be paid attention to clinically because it is the residual DTC that ultimately leads to poor prognosis
Author Response
Human tumor samples or organoids were used, which would add rigor to the study.
We added a sentence to the Discussion that we agree that in the future it will be possible to replace cancer cell lines with patient samples. We have used PDX cells in previously published work, but for the main body of this work we chose to focus on the variability of the endothelial cells on cancer cells representing a single osteoblastic cell subtype, PCS2, that represents a DTC type responsive to BMEC factors. We believe that our findings will translate to other PCS2 type DTCs, work we will plan for the future but was beyond the scope of this study.
"previous studies have shown that DTC is dormant in bone marrow. What is the state of residual tumor cells after co-culture in this 3D model? Is it in a dormant state?"
We added a paragraph to the Discussion where we comment on the role of our model to study the role of dormancy in development of therapies against DTCs that may remain distributed for years before relapse. Our data suggest that cancer clusters treated with B-CM after 14 days contain a mixture of apoptotic and dormant cells (RPPA data and western validation, TUNEL assays).
"How resistant is it to apoptosis? And how is this resistance achieved?"
Our 3D tri-culture system has allowed us to develop insights into how DTCs in the bone marrow microenvironment might respond to local factors, something not easily seen in animal models where real time observation is not possible. Our data indicates that bone marrow factors limit growth initially by a combination of pro-apoptotic and pro-dormancy activities. The dormant population would be responsible for local relapse when reactivated. We have added text to note the mixed effects to the Results and Discussion.
"Residual DTC is the most important component that should be paid attention to clinically because it is the residual DTC that ultimately leads to poor prognosis."
We agree!
Reviewer 3 Report (New Reviewer)
The study focuses on a new 3D co-culture model to study the influence of the reactive BMTE including stromal and endothelial cells on adaptive behaviors of cancer cells modeling DTCs at sites of bone metastasis. The study is well structured and comprehensively described. The hypotheses are well explained and the methodologies are accurate and suitable for the achieving of the results. The experiments are excellently designed, the results are well presented and discussed. The findings of the study provide new insights for future development of targeted therapy on prostate cancer. It would be interesting to apply the model in perspective studies using PC cells taken directly from patients suffering from prostate cancer in order to verify its the therapeutic applicability on real scenarios.
Author Response
"It would be interesting to apply the model in perspective studies using PC cells taken directly from patients suffering from prostate cancer in order to verify its the therapeutic applicability on real scenarios."
We thank the reviewer for the enthusiasm expressed for our work. We are wroking with our clinical collaborators to adapt our system to primary cells from patients. So far, these have been very difficult to get as we do not have access to a warm autopsy program and we have had little luck growing PCa cells we have gotten from bone marrow biopsies of patients with bone metastases, perhaps because they have received so much chemotherapy.
Reviewer 4 Report (New Reviewer)
Bone Marrow Endothelial Cells Increase Prostate Cancer Cell Apoptosis in 3D Triculture Model of Reactive Stroma by Sablatura et al is very interesting. However, the manuscript needs additional data/information to make it interesting to readers in the field.
Comments:
1. The authors have identified that bone marrow endothelial cells increase the apoptosis in co-culture mode. In the bone marrow environment prostate cancer cells try to interact with osteoblast and osteoclast cells and makes bone weaker, it is concern. Do authors evaluate the interaction of PCa cell with osteoblast and osteoclast cells?
2. Fig 5 shows TUNEL assay with conditioned medium (CM)-treated C4-2B clusters. Nice to see the tunel positive cells/values in graphical format.
3. Fig 6 Analysis of the phosphoproteome by reverse phase protein analysis 21 (RPPA) of PCa cells treated with conditioned medium shows the Phospho Serine 708 and 811. It will be nice to see with graphical representation of the pRb S708, S 811 and S780,
Author Response
We thank the reviewer for the enthusiasm for our work described as very interesting.
"The authors have identified that bone marrow endothelial cells increase the apoptosis in co-culture mode. In the bone marrow environment prostate cancer cells try to interact with osteoblast and osteoclast cells and makes bone weaker, it is concern. Do authors evaluate the interaction of PCa cell with osteoblast and osteoclast cells?"
We previously published our co-culture system with PCa cells and osteoblasts (PMID 26599623). This earlier work was foundational for the work here where we moved forward to create a 3D tri-culture model for PCa cells arriving to bone marrow including endothelial cells. The PCa subtype, PCS2, that we study here produces osteoblastic lesions, not osteolytic lesions in bone, as do most PCa bone metastases. We have not yet been able to culture mature osteoclasts with PCa cells in 3D, but it would be very interesting to do this in the future. Osteoclasts prefer a hard surface on which they attach and polarize, and our soft gel bone marrow mimetic system is not ideal for them. Developing a 3D system for osteoclasts and PCa cells will be a future ambitious endeavor for our team.
"Fig 5 shows TUNEL assay with conditioned medium (CM)-treated C4-2B clusters. Nice to see the tunel positive cells/values in graphical format."
Thank you for the suggestion. We have added a panel to the figure where the results were quantified and added a section to the Methods describing our use of ilastik to do this.
"Fig 6 Analysis of the phosphoproteome by reverse phase protein analysis 21 (RPPA) of PCa cells treated with conditioned medium shows the Phospho Serine 708 and 811. It will be nice to see with graphical representation of the pRb S708, S 811 and S780."
Thank you for the suggestion. We used ImageJ to quantify the band intensities in the western blots and have added this as a ratiometric change in protein levels in each treatment condition as a graphical representation under each western blot.
Round 2
Reviewer 4 Report (New Reviewer)
The quality of the manuscript is improved.
This manuscript is a resubmission of an earlier submission. The following is a list of the peer review reports and author responses from that submission.
Round 1
Reviewer 1 Report
Comments to the Author
The stated goal of this manuscript is to study the interactions among cells present in bone marrow tumor microenvironment with prostate cancer cells. The authors develop a 3D culture model using hydrogel and they study the interactions between endothelial-stromal co-cultures and endothelial-stromal-cancer tri-cultures. The results show that the microvascular endothelial cells from bone marrow induces apoptosis in prostate cancer bone metastasis through paracrine death or dormancy/quiescence-inducing signaling axis.
The article is well written and the presentation of the figures are correct. The authors have previous experience in 3D cultures that has previously been published, so the novelty of the article lies in the different effect of the different endothelial cells (according to their origin) on the tumor cell. In this way, for its publication it would be recommended that the authors review the following points.
The authors work mainly with a 4:1 ratio, although they carry out studies with other ratios (8:1, 4:1, 2:1). The authors should show that this is the approximate range that is established in the in vivo process since, depending on the range, they observe differences in the endothelial network architecture and, therefore, in the development of metastasis.
In figure 1: the legends of figure 1c and 1d are missing: (white ratio 8:1, gray ratio 4:1 and black ratio 2:1
In the caption of figure 2 it should be indicated what each color corresponds to (red??, green??) Are all the photos made with the same scale?
The authors show that treatment of endothelial cell cultures with conditioned media induces a decrease in viability at 14 days. None of the figures show the controls without treatments on the different days (only on day 1). Authors should show such controls to have this point of reference.
The authors suggest that the modulation of cell cycle regulator pRb and Forkhead box M1 (FOXM1) in prostate cancer cells is induced by BMEC conditioned media. What factors are responsible? Are these regulators responsible for the observed effects on viability and apoptosis?
Finally, the authors do not address in the discussion the fact that bone marrow endothelial cells induce death in metastatic prostate tumor cells. Finally, metastasis is established in the bone environment, so this process will be a process limited in time. Do the authors have any long-term experiments?
Reviewer 2 Report
The paper by Sablatura et al. Aimed to investigate cross-talk in a tri-culture model of osteoblastic prostate cancer cells, stromal cells and microvascular endothelial cells. This is an interesting study, however, raises some questions that are listed below:
- What is the physiological significance of the investigated cells ratio in a 3D model?
- Authors mentioned: "Tri-culture of BMEC, HS27a, and pre-clustered C4-2B induces a cell death/dormancy 299 phenotype in C4-2B". Did the authors check what actually occurring in these cells? Is that apoptosis or senescence? Cause the coculture is not the same as the usage of conditioned media.
- I would recommend confirming the induction of apoptosis using e.g. in situ TUNEL method.
- When the authors observed an increase in pRb they should also look at the senescence.